# Effects of Blueberry Consumption on Cardiovascular Health in Healthy Adults: A Cross-Over Randomised Controlled Trial

**DOI:** 10.3390/nu14132562

**Published:** 2022-06-21

**Authors:** Yueyue Wang, Jose Lara Gallegos, Crystal Haskell-Ramsay, John K. Lodge

**Affiliations:** 1Department of Applied Sciences, Faculty of Health and Life Sciences, Northumbria University, Newcastle-upon-Tyne NE1 8ST, UK; yuewang149@outlook.com (Y.W.); jose.lara@northumbria.ac.uk (J.L.G.); 2Department of Psychology, Faculty of Health and Life Sciences, Northumbria University, Newcastle-upon-Tyne NE1 8ST, UK; crystal.haskell-ramsay@northumbria.ac.uk; 3School of Human Sciences, London Metropolitan University, 166-220 Holloway Road, London N7 8DB, UK

**Keywords:** blueberry, cardiovascular, blood pressure, lipids, atherosclerosis

## Abstract

Blueberries are rich in polyphenols, and their effect on cardiovascular health, including risk factors for endothelial dysfunction and hypertension, has been investigated in interventional studies. However, the difference between blueberry treatments in varied forms for their cardiovascular-protective effect remains poorly understood. The current study assessed the effects of whole blueberry and freeze-dried blueberry powder compared to a control on cardiovascular health in young adults. A cross-over randomised controlled trial (RCT) was implemented with 1 week of treatment for three treatment groups, each followed by 1 week of wash out period. Systolic (SBP) and diastolic blood pressure (DBP), pulse wave velocity (PWV), plasma cholesterol (low-density lipoprotein cholesterol (LDL-C), high-density lipoprotein cholesterol (HDL-C), and total cholesterol) and triglyceride levels (TAG), and glucose and nitrite (NO2-) concentrations were compared following fresh blueberry, freeze-dried blueberry powder, and control treatments. Thirty-seven participants with a mean age of 25.86 ± 6.81 completed the study. No significant difference was observed among fresh blueberry, blueberry powder, and the control arm. Plasma NO2- levels were improved by 68.66% and 4.34% separately following whole blueberry and blueberry powder supplementations compared to the baseline, whereas the control supplementation reported a decrease (−9.10%), although it was not statistically significant. There were no other effects shown for SBP, DBP, total cholesterol, HDL-C, LDL-C, TAG, or glucose. No difference was shown between whole blueberry and freeze-dried blueberry powder consumption for improving cardiovascular health.

## 1. Introduction

Cardiovascular diseases (CVDs) are a major cause of death globally, resulting in an estimated 17.9 million deaths each year [1]. Raised levels of blood lipids and glucose, hypertension, and being overweight are important risk factors for CVD [2]. The development of atherosclerosis is also associated with elevated inflammatory biomarkers and uncoupled endothelial nitric oxide synthase (eNOS) [3]. Plasma nitrite (NO_2_^−^) plays an important role in the pathological state by altering endothelial function [4] and acts as an endothelium-derived relaxing factor [5]. Endothelial dysfunction in the peripheral circulation is a pathological state of the vascular endothelium (an inner layer of blood vessels) and is the manifestation of underlying atherosclerosis, CVD, or other vascular diseases [6]. The key to healthy ageing in terms of vascular change is also to maintain homeostasis of endothelial function [7].

Vitamins, minerals, polyphenols, and flavonoids from the diet have shown beneficial effects in improving endothelial function, blood pressure, blood lipid levels, and inflammatory markers [8,9,10]. However, from a comprehensive nutritional perspective, employing whole foods instead of single components as supplements in interventions is more reasonable. Well-accepted evidence from recent years indicates that normal cardiovascular health is associated with heart-healthy eating behaviours [11]. Flavonoid- and polyphenol-rich fruits, such as cherries, berries, and orange juice, supplemented in interventions have been shown to benefit cardiovascular health [12,13,14], and there is a growing interest in the impact of berry consumption on cardiovascular health.

Dark-coloured berries, such as blueberries, are rich in polyphenols (e.g., anthocyanins) and the metabolites of simple phenolic acids, which could contribute to a specific effect in reducing CVD risk factors [15]. Blueberries are very popular among consumers in the United Kingdom [16]. Whole blueberry interventions in both acute and long-term RCTs have shown improvements in blood pressure [17,18]. One recent systematic review and meta-analysis of RCTs also showed that blueberries in freeze-dried forms have been frequently studied for their cardiovascular protective effects [12]. There was also an emphasized need for additional research assessing the effect of processed fruit on health, as fruit in processed forms provides consumer options while reducing costs and food waste in the meantime [19]. Fruit powder, apart from fruit juice, could be an effective method to increase the benefits of fruit consumption, and it is useful to deliver dietary options to the public regarding fruit groups and delivery forms. However, the freeze-drying process for blueberry powder may lead to different levels of bioactives compared to the whole blueberry [20]. Therefore, it is of nutritional interest to test and compare powdered berry against whole berry in an RCT.

Ultimately, a randomised, controlled, single-blinded, counter-balanced-cross-over design was employed to evaluate the efficacy of daily whole blueberry and freeze-dried blueberry powder with equivalent polyphenol contents on cardiovascular health, including endothelial function as assessed by PWV, blood pressure, plasma lipid profile, and nitrite level following a 1-week intervention.

## 2. Materials and Methods

### 2.1. Participants

The current study adhered to the Consolidated Standards of Reporting Trials (CONSORT) guidance with the supplemented checklist (Appendix A) [21]. The study was approved by Northumbria University’s Faculty of Health and Life Sciences Ethics Committee and was registered under ClinicalTrials.gov (ID: NCT04015258). The current study aimed to recruit adults with an age range of 18–60 years old with normal (18.5–24.9 kg/m^2^) to overweight (25.0–29.9 kg/m^2^) BMI in order to include generally healthy participants with low-mild health risks. Forty participants were required based on data from two previous studies targeting cardiovascular health following 300 g of fresh frozen blueberry supplementations [17]. An effect size of d = 4.7 was calculated and indicated that a total of 35 subjects was required for detecting an effect difference in treatments separately at a two-sided 0.05 significance level with a statistical power of 0.8.

The inclusion criteria included a requirement of no relevant pre-existing medical condition/illness, such as heart disease, hypertension (≥140/90 mmHg), and neuropsychological disorders, and no current use of prescription medications (excluding contraception pills); no smoking; and a body mass index (BMI) between 18.5 and 30 kg/m^2^ (inclusive), which were all assessed on the screening visit; no current condition of migraines (>1 per month) and no visual impairment that cannot be corrected with glasses or contact lenses; being proficient in English; did not regularly consume blueberry/blueberry-containing products more than twice a week; no current participation in other clinical or nutritional interventional studies; and had not habitually used supplements within the last month (defined as more than 3 consecutive days or 4 days in total). Participants were not recruited if at least one of the criteria did not apply to them. Each participant would receive a GBP 20 shopping voucher upon their completion of the study.

### 2.2. Treatment

Fresh blueberries (*Vaccinium corymbosum* L.) were purchased from a local grocer that used the same source (Spanish origin) throughout the intervention; freeze-dried blueberry powder was purchased from Lio-Licious freeze-dried fruits (Lio-Licus, Preston, UK); a microcrystalline cellulose control was purchased from Blackburn Distributions. The powder to fresh blueberry net weight conversion was provided by the supplier.

The present study supplied either 160 g of fresh whole blueberry (four handful portion, two NHS adult portion sizes) or 20 g of freeze-dried blueberry powder (measured with provided tablespoon; equivalent to 160 g of whole fresh blueberry), or a control capsule with plant-derived fibre microcrystalline cellulose (participants were blinded that this was encapsulated blueberry extract components) to participants on separate occasions for 1 week with a 1 week washout period. Each participant was given each treatment in a random sequence. The random permutation was performed by an online randomisation tool (http://randomization.com/ accessed date: 1 November 2018), and the blinding of the control treatment was conducted by the researcher (YW) who co-ordinated the intervention.

All treatments were prepared 1 day prior to the study visit and were given to the participants in their 7 days entirety. Participants were asked to keep their blueberries refrigerated to keep them fresh. Compliance was checked on each study visit using a 1-day food diary as described below.

### 2.3. Total Polyphenol Analysis

A total polyphenol analysis of gallic acid equivalence was completed for blueberry, blueberry powder, and the control capsules using the Folin Ciocalteau reagent method [22] (Table 1).

### 2.4. Biological Sampling

For plasma separation, 20 mL of fasting venous blood was collected from participants using the BD Vacutainer™ Flashback Blood System into heparinised plasma tubes and centrifuged for 10 min at 1200× *g* relative centrifugal force to separate the plasma, which was stored in aliquots at −80 °C. The plasma was used to measure the concentrations of glucose, the lipid status (total, HDL, and LDL cholesterol and triglycerides) and the nitric oxide (NO) metabolite nitrite (NO_2_^−^).

### 2.5. Anthropometric Measurements

Participants’ age, gender, height, and weight were recorded at the screening session. Height and weight were measured using a digital scale (Seca Scales 703, Seca Ltd. Birmingham, UK), and BMI was calculated correspondingly. A 1-day food diary was collected to calculate dietary energy intake throughout the trial.

### 2.6. Whole Body Measurements of Cardiovascular Health

Systolic and diastolic blood pressures (SBP and DBP) were measured three times by a vital signs monitor (GE Carescape) with participants sitting in an upright position for 5 min. A first reading was taken but discarded. If the average reading was out of range, but the third was lower than the second reading, a fourth reading was taken [23]. This reading alone was then used as the final measurement. Readings were taken in >1 min intervals.

The pulse wave velocity of the carotid artery and radial artery (crPWV) was assessed using a SphygmoCor (ScanMed medical), and the cardiac rhythm was monitored using an electrocardiogram (ECG) pad. The readings were repeated three times.

### 2.7. Clinical Chemistry Assessment

The Randox Daytona GOD-PAP (Cat. No. GL8038) with a measuring range of 0.200–35.5 mmol/L was used to analyse glucose. The plasma lipid status was measured using the Randox Daytona GOD-PAP with a measuring range of 0.22–21.7 mmol/L (Cat. No. CH200) and 0.1–13.4 mmol/L(Cat. No. TR210) for total cholesterol and triglycerides, respectively. The Randox Daytona direct clearance method (Cat. No. CH1383) was used for measuring HDL cholesterol with a measuring range of 0.189–4.03 mmol/L. For the above analysis, a correlation of r = 0.999 against another commercially available methods was reported [24]. The LDL cholesterol level was further calculated using the Friedewald equation: LDL cholesterol (mmol/L) = total cholesterol (mmol/L) − HDL cholesterol (mmol/L) − (triglyceride/5) (mmol/L) [25]. The total amount of nitrite (NO_2_^−^) in the plasma was determined by using the chemiluminescence method via a purge system (Sievers Instruments, model NOA 280i, Boulder, CO, USA) with a repeatability of +/−5% [26]. The plasma samples used for nitrite analysis were deproteinised using ethanol to prevent foaming; the plasma was mixed with ethanol in a 2-fold dilution in microcentrifuge tubes and left to stand for 30 min, following which the tubes were centrifuged at 12,500 rpm for 5 min. The supernatant was removed for analysis. Standard solutions containing 10 nM, 50 nM, 100 nM, 1 µM, 5 µM, 10 µM, 50 µM, and 100 µM sodium nitrite (NaNO_2_) were prepared with nitrate-free deionised water and analysed to construct a calibration curve.

### 2.8. Statistical Analysis

The data for each assessed post-intervention outcome were analysed separately using linear mixed-effects models (MIXED) in SPSS statistics 26 (IBM Corp: Armonk, NY, USA). The data for each assessed outcome were checked for homogeneity of variance (Levene’s test) prior to the MIXED model. This cross-over study consisted of seven visits, including a screening session and the baseline visit, followed by the post-intervention visits. Due to the randomised treatment sequence, each post-intervention visit (three in total) consisted of the blueberry, blueberry powder, and control groups. Therefore, the effect of visit order was included as a fixed factor in the statistical analysis. The post-intervention measures were modelled, including the respective baseline values, as covariates, and the terms treatment, visit, treatment*visit, and baseline were modelled as fixed factors. The effects of sex were also explored by including sex in the model as a fixed factor, but since this did not change the effects, this term was removed from the final model. All three treatments were compared to each other. Participant was included as a random factor. A post hoc analysis used the least significant difference test (LSD) and adjusted *p* values for pairwise comparisons for treatments, visits, and interaction effects.

A paired *t*-test was used to analyse the difference in participants’ dietary total energy, carbohydrate, fat, and protein intakes across visits within each treatment arm.

### 2.9. Study Procedure

Figure 1 shows the study timeline. The 1st visit was a screening appointment; upon arrival, the participants were asked to give signed informed consent to participate. Once they had completed the consent form, they were screened for BP, general physical health (including the measurement of demographic data (age, height, and weight)), and criteria questions, as described in the inclusion criteria for an eligibility check. This screening session usually lasted approximately 2 h. All participants’ data were recorded in the pre-defined case report form. They then completed a questionnaire based upon the consumption of blueberries and related products. Once the participants met the inclusion criteria, the participants were asked to completely avoid the consumption of polyphenol-rich berries (e.g., blueberry, raspberry, blackcurrant, etc.), cherry, raisins, and vegetables (e.g., eggplant, purple cabbage, etc.) and any foods containing them. A list of food that they should refrain from taking was given to the participants. Apart from that exclusion, participants were asked to stick to their usual diet. Before they left, the participants were given a 1-day food diary to record their dietary intake on the days prior to the visit days (they were asked to bring this diary to every appointment they attended). At the end of this screening session, they were provided with urine sample collection kits and instructions on their use. They were also asked to bring the completed samples for their second urination in the morning on the day of their next visit.

The 2nd, 4th, and 6th visits were the baseline visits for each treatment. These visits took place 1 week after the last visit (ideally the same day of the week). Participants completed their urine collection on the morning of this visit and brought this sample with them to the laboratory (second urination of the morning). Their 1-day food diary questionnaire was also returned. The participants arrived at the laboratory having not consumed any food or drink other than water for 12 h prior. Upon arrival at the lab, the researcher (YW) checked that they still complied with the inclusion/exclusion criteria and were in good health and were well-rested. Changes to medication and/or illness between visits were recorded if applicable. At this session, they were required to bring the 20 mL spot urination in the morning. Then, 10 mL of their venous blood sample was taken by a suitably qualified individual for a biochemistry clinical measurement (YW) (approx. 10 min). Then, their systolic and diastolic blood pressure were measured (approx. 15 min). The next step was the non-invasive measurement of pulse wave velocity, which was taken to assess participants’ endothelial function (approx. 15 min). This was the completion of participant testing. Participants were provided with a 1-day food diary and a urine sample kit upon their departure to record their food intake of the day prior to their next visit and were reminded to bring their 2nd urination on the next visit day. The participants were instructed to take their treatment for the following 1 week and to adhere to the instructions and consume the randomly assigned supplementation. All supplementations were provided to the participants on their departure.

The procedure on the 3rd, 5th, and the 7th visits was identical to the aforementioned procedure, except that the participants were not given treatments upon their departure as they were to enter the 1-week washout period. On visit 7, all participants were asked to complete a treatment guess questionnaire in order to check their blinding status and compliance, and the voucher was given to the participants upon their completion of the last visit as compensation for the time and inconvenience.

## 3. Results

### 3.1. Participants

A total of 40 people received the interventions, and 37 people finished the trial as shown in Figure 2. Participant demographics and dietary intake at baseline are reported in Table 2.

The mean energy and macronutrient intake values pre- and post-intervention are shown in Table 3. No significant difference was found for the dietary intake within the whole blueberry and blueberry powder intervention arms.

### 3.2. Whole Body Measurements of Cardiovascular Health

The homogeneity of variance was checked in order to test the group variances prior to the interventions.

No effect of treatment was found for SBP and DBP with a covariance adjustment for baseline. Figure 3a displays the effect of the interventions on PWV. The pre- and post-intervention baseline-adjusted means of PWV and blood pressure are reported in Table 4.

### 3.3. Clinical Chemistry Assessment

Table 4 shows the pre- and post-intervention baseline-adjusted means for plasma biomarkers for each intervention group. No effects of treatments, visiting days, or the interaction effects of treatment*visit were shown for plasma triglycerides (TAG), total cholesterol, HDL cholesterol, LDL cholesterol, glucose, or the nitrite (NO_2_^−^) level with a covariance adjustment for baseline. Both blueberry supplementation and blueberry powder supplementations reported improved NO_2_^−^ levels (+68.66% and +4.34%, respectively) compared to the baselines, whereas the control supplementation reported a decrease (−9.10%). However, the difference between the blueberry treatments and the control was not statistically significant. Figure 3b displays the effect of the interventions on the plasma nitrite level.

## 4. Discussion

### 4.1. Principal Findings

To the best of our knowledge, this is the first cross-over RCT investigating the efficacy of improving endothelial function, blood pressure, and blood lipids after consuming whole blueberry (160 g) or freeze-dried blueberry powder (20 g) in a 7-day chronic intervention. We found that increasing daily blueberry consumption to 160 g/d or the equivalent blueberry powder in healthy adults for 1 week did not influence blood pressure, endothelial function, plasma lipids, glucose, or nitrite levels. The plasma NO_2_^−^ levels were improved by 68.66% and 4.34% following whole blueberry and blueberry powder supplementations, respectively, compared to the baseline, whereas the control supplementation reported a decrease (−9.10%), although it was not statistically significant (*p* = 0.148).

### 4.2. Effect of Blueberry Interventions on Cardiovascular Health

The findings on cardiovascular health are in disagreement with most non-acute blueberry interventions investigating outcomes including blood pressure (BP), lipids, endothelial function, and nitric oxide (NO) [27,28,29,30]. Even though BP is commonly evaluated as an indicator of cardiovascular function, studies exploring the ingestion of berries or their polyphenol bioactive compounds are not unequivocal on BP [12]. There have been interventions supplementing freeze-dried powder (750 mg–50 g, 4–6 months) and berry juice (100–500 mL, 2–17 weeks), including blueberry, raspberry, and grape, that have reported either no effects [27,31] or improved effects on SBP and DBP [28,32].

A trend for increased NO_2_^−^ levels was observed following the blueberry and blueberry powder interventions in the current study (*p* = 0.148). The NO synthesis system has been shown to benefit both peripheral and cerebral peripheral blood flow [33]. Anthocyanins (ACNs), rich polyphenol compounds found in blueberries, have been associated with blood pressure modulation by eliminating the synthesis of vasoconstricting molecules [34,35]. Clinical trials with dietary polyphenols have shown benefits on cardiovascular function, with one of the proposed mechanisms via the NO_3_^−^-NO_2_^−^-NO pathway [36]. Despite these, the current study found no significant effect of the interventions on SBP, DBP, or plasma nitrite. During NO synthesis, the production of NO by iNOS (an NOS isoform) has been shown to be sustained for a longer duration and found in much higher concentrations in the cell than the other isoforms of NOS [37]. Therefore, any regulatory effects on the NO systems due to blueberry polyphenols could be beneficial over a longer-term supplementation in addition to the acute (1–6 h) postprandial mechanisms [37]. As an initial rise in NO_2_^−^ level has been observed in the current study, a longer intervention duration may be needed for a beneficial observation on nitrite levels and a manifestation to blood pressure.

However, one chronic blueberry intervention did not show an improvement in endothelial function or NO levels after receiving a blueberry drink (25 g of freeze-dried powder) for 6 weeks [38]. Similar to the current study, blood samples were taken after fasting (approximately 12 h) following the blueberry ingestion, when ACNs were possibly cleared from the circulation. Currently, there are limited data of acute freeze-dried blueberry powder interventions investigating cardiovascular health. Wang et al. [12] reviewed the longer-term interventions supplementing blueberry powder (22–50 g, 6–8 weeks), and previous studies have shown inconclusive results, including the effect on NO levels [27,28,29,30,38]. Only one 6 h study supplied blueberry powder (equivalent to 240–560 g of blueberry) and demonstrated an improvement in endothelial function in healthy humans within 1–6 h after consumption [39]. Compared to blueberry powder interventions, there are several acute whole blueberry interventions (300 g, 24 h) that also showed positive results on improving peripheral cardiovascular function, but they were from the same author [17,18,40]. Nevertheless, the cardiovascular protective effects of whole fresh blueberry and blueberry powder supplementations have not been compared in a study before, despite the study duration.

The health status of the participants could also impact the study findings. Previous blueberry interventions (22–45 g of blueberry powder, 150–300 g of whole blueberry equivalent) assessing adults with high CVD risks, such as metabolic syndrome or hypertension, for 6 weeks to 6 months have reported an improvement in at least one of the outcomes (i.e., PWV, BP, lipids, and NO levels) compared to either the control or the baseline [12]. On the contrary, blueberry interventions targeting generally healthy people have reported limited changes to outcomes. An exercise intervention combined with blueberry treatment (150 g) among healthy people for 4 weeks demonstrated inconclusive evidence regarding the effect of blueberry treatment on CVD risk factors, where the HDL cholesterol level was increased in the blueberry group, but the TAG level was reduced in the control group compared to the baseline [41]. It should also be noted that McAnulty et al., [30] assessed a healthy group and observed improved endothelial function, assessed by the augmentation index (Aix), although no improvement was shown for PWV. There were also significant reductions to DBP following the blueberry supplementation compared to baseline, but an improvement was exhibited only in a subset of hypertensive participants (*n* = 9). The mean blood pressure of the overall cohort was not affected [30]. However, there was one study reporting positive findings on cardiovascular health in healthy adults following blueberry consumption. The intervention supplementing 22 g of blueberry powder daily that contained 300 mg of ACN for 1 month to low-risk healthy adults demonstrated significantly improvements in both FMD and 24 h SBP. The benefits to the cardiovascular system were shown in healthy adults following both acute and chronic blueberry consumption. Thus, in humans, populations with elevated baseline inflammatory, hypertensive, hyperlipidaemia, and endothelial (dys)function risk factors may be necessary to observe a risk-reduction from blueberry.

### 4.3. Comparison between Whole Blueberry and Blueberry Powder

In terms of the nutritional value, the blueberry powder supplemented in the current study had lower sugar content (22% less) compared to the whole blueberry, and the glucose load in the fruit matrix was shown to delay anthocyanin absorption, which could result from the competitive action of glucose and anthocyanin on the sodium-dependent glucose cotransporter SGLT-1 [42]. The blueberry skin is rich in anthocyanins and is usually kept during the freeze-drying processing of the blueberry to powder [43]. It has also been shown that fibre content is contained during the processing from blueberry to powder [44].

Despite the possible difference in the bio-accessibility and absorption of bioactives between the two treatments, there was no major effect of the interventions in cardiovascular function. No other interventional RCTs has compared whole blueberry and blueberry powder for their cardiovascular protective benefits, as far as we know. In another study comparing between blueberry and blueberry juice interventions (2 h post-consumption only), approximately 15% of metabolites were higher in intensity after whole blueberry compared with juiced (approximately 3%) blueberry [45]. In the current study, there could be differences in the bioactive and nutrient values between the freeze-dried blueberry powder and whole blueberry, but there was no treatment effect on improving endpoints, so it has not been possible to demonstrate if whole or powdered blueberry is the most effective at health promotion. The current study was part of a larger study where metabolomics were applied to help identify responders and non-responders from the interventions, which will be published elsewhere. The metabolomic profiling between two different interventional arms will also be investigated and discussed in the follow-up study.

### 4.4. Study Limitations and Implications

The prevention of CVD is the key priority for both NHS England and the government [46]. A recent science-based report of the U.S. Dietary Guidelines Advisory Committee also recommended dietary intervention for the prevention of chronic (non-communicable) diseases, including CVD [47]. However, the study here was performed on healthy people in the normal range for glucose, lipids, and other vascular measures at the baseline, rather than participants with developed disease risks.

The durations of the study interventions were chosen based on our previous systematic reviews and meta-analyses of fruit RCTs [12], which suggested that interventions spanning at least 1 week are able to significantly improve SBP and DBP and thus to benefit vascular function. Supplementing for 4 days has also been observed to show distinctive metabolomic changes with a dietary intervention [48].

The wash-out period was implemented to eliminate the carry-over effect of different treatment arms in a cross-over study since each participant received all the treatments in a different orders [49]. The wash-out period is usually the length of five times the half-life of the treatment [50]. Based upon the data showing a maximum half-life for major fruit polyphenols of 28.1 h [51,52], 1 week was deemed to be sufficient. We also considered using a 1 week intervention, as it was more convenient for participants in keeping track of visiting times to the clinical investigation centre on the same weekday/weekend day every week. The participants’ burden is considered in a clinical trial to minimise the risk of dropout [53]. Previous RCTs also implemented a longer duration of dietary intervention, whereas a more realistic intervention duration should be taken into account to assess their impact over a shorter period and also assure good compliance from participants [54]. Here, all assessed endpoints for study participants were in the normal range, which could account for the absence of the improvement in assessed cardiovascular health. Furthermore, the dosage of blueberries was equivalent to that recommended by NHS [55], which was expected to augment the efficacy within a shorter term.

The total polyphenol content, but not anthocyanins, in blueberry and freeze-dried blueberry powder were measured and reported in this study. A validation of bioavailability may be necessary. Nevertheless, the excreted anthocyanin levels usually are very low [56] and they are metabolised to low-molecular-weight phenolic acids, and several other factors affecting the response to the interventions should be considered. Potential covariances of age, BMI, energy intake, socioeconomic status, and physical activity were not included in the analyses, but there were mostly young participants (25.86 ± 6.81 years old) of white ethnicity (*n* = 31), and all were healthy with a BMI under 30 kg/m^2^ and non-smoking status during the trial. Because the current study aimed to recruit healthy participants in the normal physiological range, the treatment effect might be different in a cohort with disease risks. The study was not powered to test the effect of BMI variability, and a wider range of BMI would be needed to explore this further. Although there were predominantly (65%) female participants in the study and the pluripotent effect of oestrogen on cardiovascular health should be considered, including the influence on coagulation, blood lipid profile, endothelium, inflammation, and cell adhesion [57], including sex, in the model did not alter the effects of the treatments.

## 5. Conclusions

The study showed no effect of consuming either blueberry or blueberry powder daily, containing 220.48 mg and 288.43 mg of total polyphenol contents, respectively, for 1 week on blood pressure, pulse wave velocity, lipid profile, glucose control, and plasma nitrites levels. This suggested no difference between whole blueberry and freeze-dried blueberry powder consumption for improving cardiovascular health within this time frame and dosage. However, the potential effect on NO_2_^−^ levels suggested a longer intervention duration might be needed. Controlling for other covariates at the baseline, including gender, BMI, age group, and ethnicity, and the investigation of the bioavailability for different forms of blueberry interventions are also suggested for future studies to better understand how each co-factor contributes to the vascular-protective effect following a dietary blueberry intervention.

## Figures and Tables

**Figure 1 nutrients-14-02562-f001:**
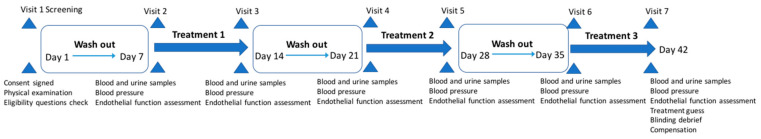
Outline of the study protocol.

**Figure 2 nutrients-14-02562-f002:**
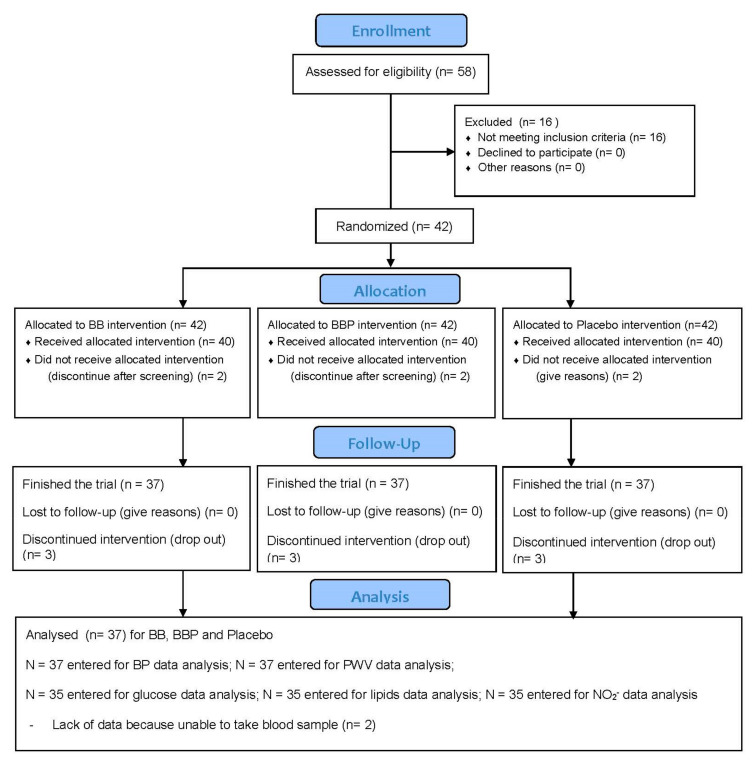
CONSORT flow diagram.

**Figure 3 nutrients-14-02562-f003:**
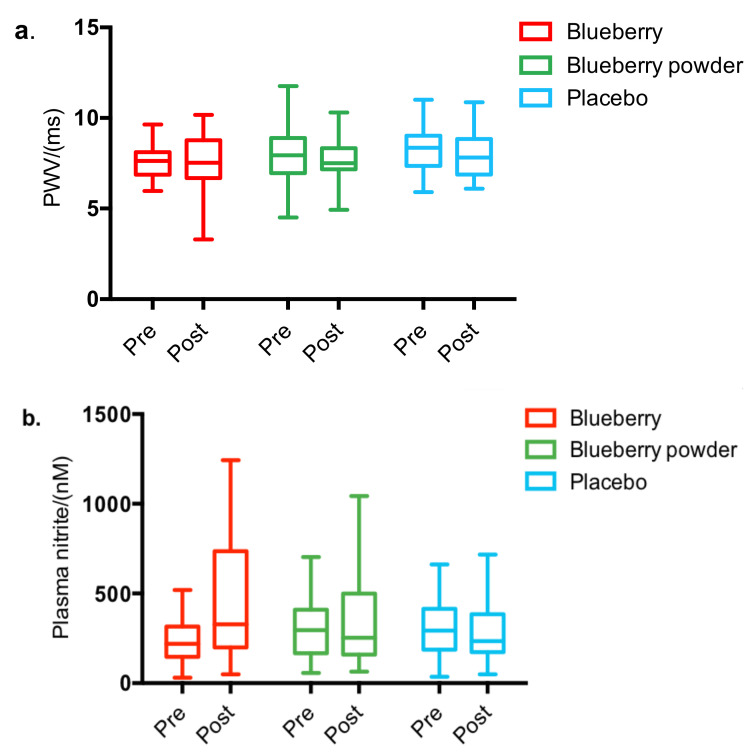
The effect of blueberry interventions on selected cardiovascular function endpoints, including (**a**) PWV and (**b**) plasma nitrite levels.

**Table 1 nutrients-14-02562-t001:** Nutrient composition of freeze-dried blueberry powder and control compared with whole blueberry.

		Blueberry (per 160 g) ^1^	Blueberry Powder (per 20 g) ^2^	Control (per 1 g) ^3^
Energy (kcal)		120.00	69.20	0.00
Fat (g)		0.32	1.30	0.00
From Saturates (g)		0.00	0.08	0.00
Total Carbohydrates (g)		23.20	13.90	1.00
From Sugars (g)		22.40	7.78	0.00
Protein (g)		0.96	1.40	0.00
Total Polyphenol Analysis (TPC) ^4^	Gallic Acid Equivalence (mg/d) ^4^	220.48	288.43	0.00

^1^ US department of Agriculture National Nutrient Database for Standard Reference. ^2^ Lio-Licious freeze-dried blueberry powder information. ^3^ Blackburn Distribution information. ^4^ Analysed by the researcher (YW) at Northumbria University using Folin Ciocalteau reagent method.

**Table 2 nutrients-14-02562-t002:** Participant demographics and dietary intake.

Variables	Value ^1^
Age (years)	25.86 ± 6.81
BMI (kg/m^2^)	23.15 ± 3.12
Gender	13 male, 24 female
Ethnicity	1 Black
	2 Indian Asian
	3 Chinese Asian
	31 White European
Fruit and vegetable intake (portions/day)	2.13 ± 0.85
Berry intake (portions/day) ^2^	0.06 ± 1.61

^1^: Data are expressed as means ± SD; BMI: body mass index. ^2^: One portion size is equal to 80 g.

**Table 3 nutrients-14-02562-t003:** Participant dietary intake pre- and post-interventions.

	Pre				Post			
	Blueberry	Blueberry Powder	Placebo	Significance	Blueberry	Blueberry Powder	Placebo	Significance
Energy (kcal) ^a^	1580.697 (425.250)	1485.929 (406.151)	1487.353 (453.500)	*p* ≥ 0.581	1583.227 (424.395)	1490.961 (409.881)	1489.849 (453.103)	*p* ≥ 0.591
Total Carbohydrates (g)	170.112 (50.693)	161.009 (57.216)	163.453 (68.156)	*p* ≥ 0.806	170.372 (51.247)	177.028 (86.868)	163.619 (68.344)	*p* ≥ 0.735
Fat (g)	57.512 (22.635)	57.926 (22.238)	55.923 (24.775)	*p* ≥ 0.932	58.650 (22.168)	59.163 (21.634)	57.388 (25.154)	*p* ≥ 0.948
Protein (g)	79.629 (48.500)	65.235 (34.302)	71.323 (37.794)	*p* ≥ 0.346	80.098 (48.638)	63.435 (24.886)	71.989 (37.826)	*p* ≥ 0.206

^a^: Data are expressed as means ± SD.

**Table 4 nutrients-14-02562-t004:** Change in PWV, blood pressure, and plasma biomarker levels from pre- to post-intervention by the blueberry intervention groups ^1^.

	Blueberry Intervention			Blueberry Powder Intervention			Placebo Intervention			Effects ^2^
	Pre	Post	△	Pre	Post	△	Pre	Post	△	
PWV, m/s	7.630 (0.153)	7.443 (0.239)	−0.187	7.993 (0.258)	7.526 (0.234)	−0.467	8.265 (0.209)	7.891 (0.244)	−0.374	*p* = 0.567
BP, mmHg										
Systolic	108.727 (1.411)	108.704 (1.815)	−0.023	110.278 (1.711)	111.395 (1.806)	1.117	109.314 (1.691)	109.732 (1.822)	0.418	*p* = 0.540
Diastolic	64.059 (1.440)	63.369 (1.398)	−0.690	64.333 (1.482)	64.048 (1.388)	−0.285	63.676 (1.395)	64.626 (1.406)	0.950	*p* = 0.366
Plasma biomarkers										
TAG	0.820 (0.053)	0.840 (0.075)	0.020	0.894 (0.059)	0.918 (0.071)	0.024	0.825 (0.061)	0.880 (0.074)	0.055	*p* = 0.960
Total cholesterol	4.302 (0.171)	4.557 (0.136)	0.255	4.567 (0.195)	4.533 (0.125)	−0.034	4.509 (0.161)	4.324 (0.134)	−0.185	*p* = 0.402
LDL-C	1.389 (0.147)	2.829 (0.136)	1.440	2.875 (0.138)	2.911 (0.128)	0.036	2.858 (0.128)	2.688 (0.136)	−0.170	*p =* 0.171
HDL-C	2.749 (0.064)	1.439 (0.068)	−1.310	1.514 (0.093)	1.443 (0.066)	−0.071	1.487 (0.080)	1.516 (0.068)	0.029	*p =* 0.989
Glucose	5.755 (0.137)	5.952 (0.155)	0.197	5.818 (0.12)	5.788 (0.145)	−0.030	5.854 (0.171)	5.625 (0.155)	−0.229	*p =* 0.659
Nitrite/NO_2_^−^, nM	236.690 (21.086)	399.190 (47.030)	155.733	310.371 (31.311)	323.84 (45.19)	27.368	305.95 (32.708)	278.12 (45.249)	−74.967	*p =* 0.184

**△** Changed scores compared to the baseline: (post-intervention—pre-intervention). ^1^ Values are baseline-adjusted means ± SEs unless otherwise indicated; ^2^ Only *p* values for treatment effect were reported; Adjusted for pairwise comparison: least significant difference (LSD) *p* ≤ 0.0167 for all pairwise comparison significance and effects observed. PWV, pulse wave velocity; BP, blood pressure; TAG: triglyceride; LDL-C: Low-density lipoprotein cholesterol: HDL: high-density lipoprotein cholesterol; nM, 10^−9^ mol/L.

## Data Availability

The data presented in this study are available on request from the corresponding author. The data are not publicly available due to ethical restrictions.

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
