# Peer review of "Effects of Blueberry Consumption on Cardiovascular Health in Healthy Adults: A Cross-Over Randomised Controlled Trial"

_nutrients, 2022, doi:10.3390/nu14132562_

Round 1

Reviewer 1 Report

I think it's a good job.

2.1. Participants: they should consider that the variety of BMI of the person, represents a variability, since the effect of blueberries (whole and powder) not only depends on consumption, but also on the habits of the participants, their diet, among others. Perhaps because of this, only in some parameters did they find effect, but not in all those that were measured. It is suggested that this part be better supported.

2.2. Treatments: Did you consider using a lens? A person who did not consume blueberries. Is it clearly placed in the methodology?

 It is necessary that in several described methodologies it is considered to add references that support.

 Improve the resolution of Figure 1.

 It should be considered that the process of obtaining blueberries powder, with loss in their properties, which are reflected in the effect that was had with the intake.

 Improve conclusions

Author Response

We thank the reviewers for their comments and our corrections appear as track changes in the revised manuscript, and are detailed here:

Reviewer 1 (Comments to the Author):

I think it's a good job.

2.1. Participants: they should consider that the variety of BMI of the person, represents a variability, since the effect of blueberries (whole and powder) not only depends on consumption, but also on the habits of the participants, their diet, among others. Perhaps because of this, only in some parameters did they find effect, but not in all those that were measured. It is suggested that this part be better supported.

Response: A discussion of BMI has been added to page 14 lines 407-414.

The dietary energy intake data of participants were collected from the food diary throughout the study and analysed pre- and post - interventions but no significant differences were found for dietary intake between and within the whole blueberry and blueberry powder intervention arms. The content was added in Methods to lines 139 – 140 on page 4, lines 185 – 186 on page 5, results were added to 240 – 243 on page 7 and as Table 3.

2.2. Treatments: Did you consider using a lens? A person who did not consume blueberries. Is it clearly placed in the methodology?

Response:The study recruited participants that consume no or low blueberry intake, at stated in lines 91 -  92 on page 2 the inclusion criteria: “did not regularly consume blueberry/blueberry-contained products more than twice a week” and their berry intake frequency data at baseline were collected from the questionnaire during screening, where their fruit and vegetables intake (portions/day) was 2.13±0.85 portions/day and berry intake was 0.06±1.61 portions/day as shown in Table 2, so the participants recruited in this study did not consume blueberries compared to the supplemented dosage.

 It is necessary that in several described methodologies it is considered to add references that support.

Response:References are added to lines 145, 158, 163 as references 23, 24, 26 on page 4.

 Improve the resolution of Figure 1.

Response:Resolution is improved for Figure 1.

It should be considered that the process of obtaining blueberries powder, with loss in their properties, which are reflected in the effect that was had with the intake.

Response:The difference between the whole blueberry and freeze-dried blueberry powder was addressed in Discussion ‘Comparison between whole blueberry and blueberry powder’ on lines 354– 375 on page 13. The difference in terms of the anthocyanins bio-accessibility and other nutritional values that could affect the bio-accessibility leading to the reflected treatment effects due to the freeze-drying processing have been discussed.

Improve conclusions

Response:The conclusion is improved as shown on lines 420 - 430 on page 14.

We hope that our revised manuscript, based on the reviewers’ comments, is now acceptable for publication.

Yours sincerely,

Reviewer 2 Report

This is a well-written manuscript describing outcomes of a randomized controlled trial (RCT) aimed at assessing the effects of whole blueberry and freeze-dried blueberry powder on cardiovascular health in young adults.  While the impact of blueberries on cardiovascular (CV) health are well known, the difference between blueberry treatments in varied forms for their cardiovascular protective effects remains poorly understood. 

There are several major concerns that are apparent as detailed here:

Is only a single week of treatment sufficient to induce changes, or should the length of treatment be extended in order to replicate a more chronic dosing as related to diet?  Also related to timing, what is the reasoning behind a one-week washout period following dietary intervention?  The washout periods are shown in Figure 1, but scientific reasoning and explanation of their use is not clear.  With timing of these experimental approaches, perhaps the lack noticeable differences between whole blueberry and freeze-dried blueberry powder consumption for improving cardiovascular health are not related to the particular forms of dietary intake but rather may be related to limitations associated with timing in the experimental design. 

Also related to timing, when were the biological samples and anthropometric readings made following the washout period?  If this timing was not appropriate (ie., if these measurements were taken too long after the study ended), then perhaps some effects could have been lost and not apparent in the analysis.

Considering possible differences in the bio-accessibility and absorption of bioactive metabolites between the different blueberry dietary interventions, for precise science it would be recommended to obtain some validation of the bioavailability of key metabolites of each regimen.

Dis-aggregation of data by sex was not performed, yet given the full cohort (13 male, 24 female) there should be ample size and power with which to analyze sex as a covariate (this would also address, given the age of these participants, the influence of female hormones on CV health as a possible adjuvant to blueberry ingestion).

Separation of whole blueberries into individual components would help identify key chemicals/proteins involved as well as their mechanisms at play; however, there is value in using whole blueberries as opposed to individualizing separate components in blueberries, which then may not result in the same beneficial effects (the sum is greater that the individual components added up).  The authors do address individual components of blueberries as well as the integrity of using intact blueberries.

The ultimate conclusion by these authors is that ‘No difference was shown between whole blueberry and freeze-dried blueberry powder consumption for improving cardiovascular health.’  Despite the lack of significant differences observed, there is value in reporting ‘negative data’. 

Author Response

We thank the reviewers for their comments and our corrections appear as track changes in the revised manuscript, and are detailed here:

Reviewer 2 (Comments to the Author):

This is a well-written manuscript describing outcomes of a randomized controlled trial (RCT) aimed at assessing the effects of whole blueberry and freeze-dried blueberry powder on cardiovascular health in young adults.  While the impact of blueberries on cardiovascular (CV) health are well known, the difference between blueberry treatments in varied forms for their cardiovascular protective effects remains poorly understood. 

There are several major concerns that are apparent as detailed here:

Is only a single week of treatment sufficient to induce changes, or should the length of treatment be extended in order to replicate a more chronic dosing as related to diet? 

Response:Justification of using the washout period and 1 week study duration was added to lines 383 - 395 on page 14. The limitations of the study length in terms of the potential treatment effect of plasma NO2- levels are discussed in lines 310 – 312 on page 12. 

Also related to timing, what is the reasoning behind a one-week washout period following dietary intervention?  The washout periods are shown in Figure 1, but scientific reasoning and explanation of their use is not clear.  With timing of these experimental approaches, perhaps the lack noticeable differences between whole blueberry and freeze-dried blueberry powder consumption for improving cardiovascular health are not related to the particular forms of dietary intake but rather may be related to limitations associated with timing in the experimental design. 

Response: Justification of using the washout period and 1 week study duration was added to lines 383 - 395 on page 14.

Also related to timing, when were the biological samples and anthropometric readings made following the washout period?  If this timing was not appropriate (ie., if these measurements were taken too long after the study ended), then perhaps some effects could have been lost and not apparent in the analysis.

Response:Collections of biological samples and the whole-body measurements following the washout period were made as baseline measurements for each treatment group to validate that there was no carryover treatment effect, not for assessing the treatment effect. The timing and duration for collecting bio samples and whole body measurements were kept consistent after either washout period or treatment period and normally be in the morning after an over-night fasting as described in lines 207– 210 on page 5. For treatment assessment on visit 3, 5, 7 as described in lines 227 – 232 on page 5, the vascular assessment procedure took approx. 40 minutes, the timing for blood sample collection has been added to lines 217 on page 5. Overall, these are not lengthy measurements in terms of the timing on the assessment visit day.

Considering possible differences in the bio-accessibility and absorption of bioactive metabolites between the different blueberry dietary interventions, for precise science it would be recommended to obtain some validation of the bioavailability of key metabolites of each regimen.

Response: The current study was part of a larger study where metabolomics was applied to help identify responders and non-responders from the interventions, which will be published elsewhere. Explanation was added to lines 372 – 375 on page 13. The metabolomic profiling between 2 different interventional arms will also be investigated and discussed in the follow-up study and the bioavailability validations are not usually made for chronic interventions but we did see the presence of anthocyanin metabolites consistent with blueberry interventions.

Dis-aggregation of data by sex was not performed, yet given the full cohort (13 male, 24 female) there should be sample size and power with which to analyze sex as a covariate (this would also address, given the age of these participants, the influence of female hormones on CV health as a possible adjuvant to blueberry ingestion).

Response: We explored the effects of sex by including sex as a fixed factor within the model. As this didn’t change the results, sex was excluded from the final model. This has been added to Page 4, line 180-182. The influence of female hormones on CV health along with discussion of other potential covariance of participants were added to lines 414 - 418 in discussion on page 14 as suggested.

Separation of whole blueberries into individual components would help identify key chemicals/proteins involved as well as their mechanisms at play; however, there is value in using whole blueberries as opposed to individualizing separate components in blueberries, which then may not result in the same beneficial effects (the sum is greater that the individual components added up).  The authors do address individual components of blueberries as well as the integrity of using intact blueberries.

Response: The difference between the whole blueberry and freeze-dried blueberry powder was addressed in Discussion to lines 354 - 375 on page 13. The difference in terms of anthocyanin bio-accessibility and other nutritional values that could affect the bio-accessibility due to the freeze-drying processing have been discussed.

The ultimate conclusion by these authors is that ‘No difference was shown between whole blueberry and freeze-dried blueberry powder consumption for improving cardiovascular health.’  Despite the lack of significant differences observed, there is value in reporting ‘negative data’. 

Response: The conclusion is improved as shown on lines 420 – 430 on pages 14.

We hope that our revised manuscript, based on the reviewers comments, is now acceptable for publication.

Yours sincerely,
